# The function of Anr in the differential effects of oxygen levels on biofilm development and nitrogenase performance in *Pseudomonas stutzeri* A1501

Noureen Rasheed[1,2,3], Xize Ali[4,5,6], Shahab -ud-din[5], Irfan Khan[2], Arsalan Hassan[3], Muhammad Ali Rasheed [ID][4,6]*

1 Department of Biochemistry, University of Karachi, Karachi, Pakistan, 2 Center for Regenerative Medicine and Stem Cell Research, The Aga Khan University, Karachi, Pakistan, 3 Department of Microbiology, Basic Medical Science Institute (BMSI), Jinnah Post Graduate Medical Center (JPMC), Karachi, Pakistan, 4 Department of Microbiology, University of Karachi, Karachi, Pakistan, 5 Department of Chemistry, University of Karachi, Karachi, Pakistan, 6 Department of Microbiology, Federal Urdu University of Arts, Sciences & Technology, Karachi, Pakistan

* malirasheed2004@gmail.com

## Abstract

*Pseudomonas stutzeri* A1501 exhibits a rare and notable trait: nitrogenase activity, which functions under microaerophilic conditions with limited oxygen availability. Optimal biofilm formation occurs in minimal media under nitrogen-depleted conditions. Anr is a transcription regulator with a widespread influence that accelerates the process of biofilm development. The lack of Anr adversely affects nitrogen fixation by regulating the activity of *nif*A, *nif*H, and *ntr*C. The *anr* insertion mutant significantly reduced the nitrogenase activity. Nitrogenase activity demonstrated considerable variation at different oxygen concentrations. The quantitative reverse transcription polymerase chain reaction (qRT-PCR) demonstrated a reduction in the expression of *nif* island genes during nitrogen fixation in the absence of *anr* gene. The discovery revealed that different oxygen levels in the environment significantly influence nitrogenase activity and biofilm formation. The qRT-PCR investigation demonstrated an upregulation of *nar*L gene expression during biofilm formation, suggesting that reduced oxygen levels initiate a signaling cascade that activates the *anr* gene. This influences both the expression of RpoS (sigma factor) genes and the process of biofilm formation.

## Introduction

*Pseudomonas stutzeri* A1501 is a type of bacterium that lives freely in the environment and can break down dead material. It is known for its ability to change its genes and perform various metabolic functions, especially its skill in denitrifying, which

**Data availability statement:** All data are present in the manuscript, and its supplementary files.

**Funding:** The author(s) received no specific funding for this work.

**Competing interests:** No authors have competing interests.

means it can turn nitrates into nitrogen gas, an important part of the nitrogen cycle [1]. The bacterium's capacity to flourish in both aerobic and anaerobic environments highlights its versatility and ecological significance. Specific strains of *P. stutzeri* exhibit bioremediation skills, proficiently decomposing contaminants including hydrocarbons and heavy metals, highlighting their potential in environmental remediation efforts. The species *P. stutzeri* A1501 is commonly isolated from soil, water, and sediments, with certain strains also associated with plants and animals, demonstrating its extensive environmental distribution. Its genetic plasticity facilitates swift responses to fluctuating environmental conditions, rendering it an exemplary organism for investigating bacterial evolution and adaptability. The quorum sensing mechanisms in *Pseudomonas* species, which facilitate cell-to-cell communication through signaling molecules, govern numerous physiological processes, including the production of virulence factors, responses to oxidative stress, antibiotic tolerance, biofilm formation, and motility [2].

The complicated system that helps bacteria adjust to different environments usually includes complex regulators and signaling molecules. In *Pseudomonas stutzeri* A1501, the anaerobic regulator Anr, the global regulator GacA, and the alternative sigma factor RpoS are identified as key regulatory proteins [3]. These regulatory proteins manage different biological functions, such as breaking down nitrates, forming biofilms, and producing harmful factors, which affect how well the bacterium survives and succeeds in its environment. The complex control by Anr, GacA, and RpoS shows how well *P. stutzeri* A1501 can adjust to different environments and demonstrates the advanced ways bacteria survive in various places.

The anaerobic regulator Anr, which is part of a group of proteins that control gene activity, is the main controller of gene expression when there is little oxygen in *P. stutzeri* A1501. Anr can directly detect how much oxygen is present using its iron-sulfur cluster, and when it finds low oxygen levels, it turns on the genes needed for anaerobic respiration, such as those for nitrate reductase. Furthermore, ANR can suppress the expression of genes linked to aerobic respiration to improve metabolic efficiency under anaerobic conditions. Anr controls the activity of genes related to both aerobic and anaerobic metabolism, helping *P. stutzeri* A1501 to effectively use available electron acceptors and keep its energy levels stable. The influence of Anr goes beyond just basic metabolism; it also affects genes involved in biofilm formation and quorum sensing, indicating that it plays a larger role in helping bacteria adapt to low-oxygen environments.

The GacA protein, a response regulator in the two-component GacS/GacA system, functions as a global regulator of gene expression in *P. stutzeri* A1501, influencing multiple cellular processes, such as secondary metabolite synthesis, biofilm formation, and virulence. The GacS/GacA system is triggered by various environmental signals, like how many cells are present and the availability of nutrients, which leads to the activation of GacA and the start of target gene expression. GacA controls the production of small non-coding RNAs (sRNAs), which then affect how target mRNAs are translated, helping to adjust gene expression after transcription. This complex system helps *P. stutzeri* A1501 quickly adapt to changing environmental conditions

and manage important life processes. The way GacA controls biofilm formation is crucial for the bacterium's survival in nature because biofilms help protect against environmental threats and enhance nutrient absorption. Small RNAs are potential regulators of biofilm formation; however, their biological functions within the cell are not yet fully understood [4].

RpoS, a special protein that helps control other genes such as catalases (*katE, katB*), superoxide dismutases (*sodB*), peroxidases, and osmoprotectant transporters (*proU, opu* family), manages the response to stress, survival when the bacteria are not growing, and how harmful *P. stutzeri* A1501 can be. Nutrient deprivation, oxidative stress, and increased cell density typically trigger RpoS, enabling bacteria to adapt to harsh environmental conditions. RpoS-dependent gene expression is regulated at multiple levels, including transcription, translation, and protein stability, thereby guaranteeing rigorous control over the stress response. RpoS encourages the production of genes that help restore DNA, resist oxidative stress, and change the cell wall, which improves the bacterium's ability to handle tough environmental challenges. The RpoS regulon includes genes that help bacteria form biofilms and communicate with each other, showing its importance in helping bacteria adapt when they are not growing actively. RpoS is acknowledged to affect antimicrobial resistance to particular bacteriocins [5]. The complicated system controlled by RpoS shows how important stress response systems are for bacteria to survive and thrive in changing environments.

The Anr protein, which responds to oxygen levels, is crucial for controlling the genes that help the organism survive without oxygen, including those for denitrification, as oxygen levels change. The importance of ANR is that it can detect when there is not enough oxygen and adjust gene activity to help the organism use anaerobic respiration and fix nitrogen when oxygen levels drop. This regulatory protein, belonging to the Crp/Fnr superfamily, acts as a molecular switch, enabling the bacterium to transition between aerobic and anaerobic metabolic pathways [6]. The presence of substantial quantities of genes involved in dissimilatory nitrate reduction to ammonium and assimilatory nitrite reduction highlights the potential for microorganisms to ensure the retention of bioavailable inorganic nitrogen in an ecosystem [7]. Research indicates that some harmful bacteria can separate the process of reducing nitrogen oxides from moving protons and making ATP or $NH_4$, but they can still connect it to the oxidation of quinol and NADH.

Certain bacteria and archaea predominantly carry out nitrogen fixation, a crucial biological process, underscoring their significance in making atmospheric nitrogen bioavailable [8]. Only bacteria and archaea distribute nitrogenase, underscoring their unique ability to carry out this crucial process [9]. The enzyme nitrogenase, which changes atmospheric nitrogen into ammonia, is made up of complex proteins that include iron, molybdenum, and sulfur. This enzyme complex is very sensitive to oxygen, so it needs low-oxygen conditions or special protections to work well, making it important to study how organisms like *P. stutzeri* A1501 carry out nitrogen fixation when oxygen levels change. Many nitrogen-fixing bacteria engage in symbiotic relationships with plants, where the plant provides a microaerobic environment and carbohydrates to support bacterial metabolism, while the bacteria supply fixed nitrogen to the plant. Siderophores, produced by aerobic and facultative anaerobic bacteria and fungi, play a role in iron transport [10].

Nitrate assimilation, which is the process of changing nitrate into nitrite, is an important step that is controlled by nitrate reductase, an enzyme that helps this process happen. Depending on the environmental conditions and specific enzymes present, the subsequent reduction of nitrite can result in either ammonium production or denitrification [11]. The regulation of nitrogen metabolism in *P. stutzeri* A1501 is carefully controlled, with the Anr protein serving as a key regulator that adjusts the activity of genes related to both nitrogen fixation and denitrification, depending on how much oxygen is available. The Anr protein influences the activity of genes that produce nitrogenase, making sure nitrogen fixation happens effectively when there is little oxygen, while also controlling the genes for denitrification to support energy production when oxygen is scarce. The way oxygen levels, regulation, nitrogenase activity, and denitrification processes.

The interaction between Anr, GacA, and RpoS in *P. stutzeri* A1501 shows how complicated bacterial control systems are and how important it is to combine different environmental signals to trigger the right responses in the bacteria. These regulators can directly or indirectly affect gene expression in response to complex environmental inputs. Anr and GacA might work together to control the genes involved in biofilm formation when there is no oxygen, while RpoS could

influence the genes that help the bacteria respond to stress caused by a lack of nutrients or oxidative stress. This communication regulates several phenotypes in bacteria [12]. Understanding how Anr, GacA, and RpoS work together is important for figuring out how *P. stutzeri* A1501 adapts to different environmental conditions and stays viable in its ecosystem. The collection comprises numerous *Pseudomonas* strains from various organizations, in addition to several reference *pseudomonas*. The neighbor-joining tree was generated utilizing MEGA 7.0 software. We evaluated the reliability of the inferred tree by conducting 1000 bootstrap trials to determine the nodal supports. The scale bar denotes the frequency of substitutions per site. (**Fig 1**) The phylogenetic analysis shows that the *anr* gene in *Pseudomonas* species is closely related to genes in other groups, like Cellvibrio, Halomonas, and Rhodospirillum. The quorum sensing system in *P. aeruginosa* can easily adapt and respond to outside stress signals. This intricate hierarchy consists of at least four interconnected signaling channels [13]. The role of regulatory cascades in gene expression during stress indicates that genes that respond to oxidative stress can be grouped into different categories.

## Method and materials

### Bacterial strains, culture media, plasmids and growth conditions

Strains, plasmids, and mutant strains are listed in Table 1 (a) and (b). The *P. stutzeri* A1501 strain was at 30°C. The organism was cultured at 30 °C with continuous agitation at 220 revolutions per minute for one night. The antibiotics were employed at concentrations of 50 µg/mL for tetracycline (Tc) and 50 µg/mL for kanamycin (Km). The enzymes necessary for DNA modification and endonuclease activity were obtained from New England Biolabs S1 Table. The primers used for the constructs are mentioned in S2 Table. There is an illustration in S1 Fig of the replacement of *anr* gene with the plasmid in the genome of *P. stutzeri* A1501. The confirmation band in the genome is present in S2 Fig.

### Growth curve analysis

The growth curve analysis was conducted using Luria Bertani (LB) medium and medium-K Table 2 (a) and (b). Following the initial $OD_{600}$ of 0.1, the culture was incubated at 30°C in 20 mL of media and shaken at 220 rpm. Growth changes were recorded on the spectrophotometer U-3010 at two-hour intervals.

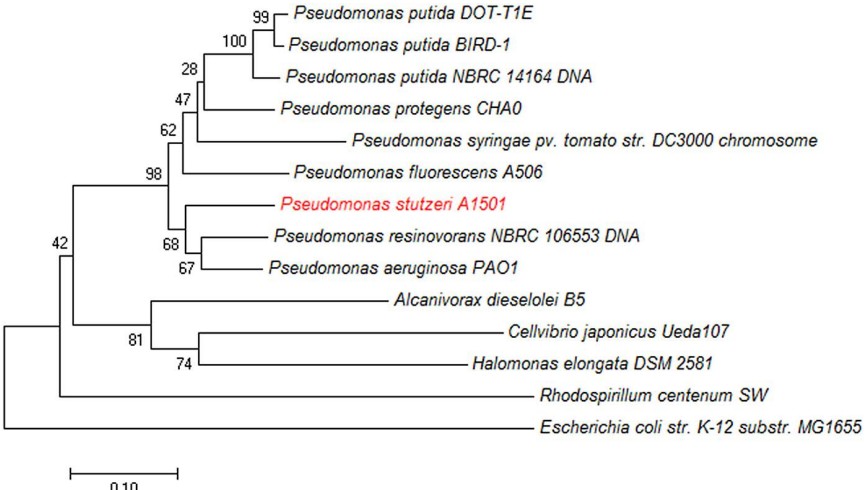

**Fig 1. Phylogenetic tree of A1501 *anr* gene.** Bootstrap value on 1000 replication is listed as a percentage at the branching point bar, 0.10 substitutions per position.

## Oxidative stress analysis

The susceptibility of *Pseudomonas stutzeri* A1501 and its derivative cells to $H_2O_2$ was assessed. The strains were cultured overnight in LB broth at a temperature of 30°C. They were then transferred into new LB broth until reaching an optical density ($OD_{600}$) of 0.1. The soup was incubated at 30°C with shaking at 220 rpm for approximately 3-hours, allowing the $OD_{600}$ to reach 0.6. Subsequently, a 12 mM concentration of hydrogen peroxide ($H_2O_2$) was introduced into the media. The dosage was selected as it was used in the similar study for the oxidative stress study [18]. The culture was cultivated at a temperature of 30°C and a speed of 220 revolutions per minute for a duration of 10 minutes. OD-standardized cultures were diluted in a series of 10-fold dilutions and then spotted on LB plates. The plates were placed in an incubator set at a temperature of 30°C for a duration of 24 hours before counting the number of colonies. The survival rate was quantified as the proportion of colonies in the treated samples relative to the untreated A1501 sample, which served as the control.

## Nitrogenase activity assays

We evaluated the nitrogenase activity by incubating bacterial suspensions in a nitrogen-free minimum medium with an optical density of 0.1 at 30°C. The incubation occurred in an argon environment with fluctuating oxygen concentrations

**Table 1. (a) Shows the plasmids used in the following study. (b) Shows the strains used in the following study.**

| Plasmids | Description | Reference |
|---|---|---|
| pRK-2013 | Helper plasmid for conjugation into *P. stutzeri* A1501, Kanamycin (Km$^r$) | [14] |
| pK-18 mob insertional plasmid | Mobilizable plasmid containing an *Escherichia coli* origin of replication, Kanamycin (Km$^r$) | [15] |
| pLAFR-3 | Mobilizable vector, tetracycline (Tc$^r$) | [16] |
| pET52(b) | Ampicillin resistance (amp$^r$) | Novogene |
| **Strains** | **Description** | **Reference** |
| *P. stutzeri* A1501 | Wild type strain, Chinese culture | [17] |
| Δ*anr* | Δ*anr*-Km insertion mutant, Km$^r$ | Used in this study |
| Δ*anr*+*anr*+pLAFR | Δ*anr*-Kmr+pLFAR-Tc | Used in this study |

**Table 2. Medium compositions. a) LB medium: 1 L, pH 7.0 (solid medium containing 1.5% agar powder). b) A15 Minimal K-Medium (Medium K): 1 L, pH 6.8 (1.5% agar powder in solid medium).**

| Composition | Weight (g) |
|---|---|
| Yeast extract | 5 |
| Tryptone | 10 |
| NaCl | 10 |
| **Composition** | **Weight (g)** |
| $KH_2PO_4$ | 0.4 |
| $K_2HPO_4$ | 0.1 |
| NaCl | 0.1 |
| $MgSO_4 \cdot 7H_2O$ | 0.2 |
| $MnSO_4 \cdot H_2O$ | 0.01 |
| $Fe_2(SO_4)_3 \cdot H_2O$ | 0.01 |
| $Na_2MoO_4 \cdot H_2O$ | 0.01 |
| $C_3H_5NaO_3$ | 6 ml |
| $(NH_4)_2SO_4$ | 0.4 |

and 10% acetylene within a 110-mL vial. The suspension was agitated at 220 rpm, adhering to the methodology established by Desnoues N [19]. We set the oxygen level to 0.5, added acetylene, and regularly took gas samples to measure ethylene production using gas chromatography. The nitrogenase activity was tested without oxygen by growing bacteria in a basic lactate solution that didn't have nitrogen but contained 1 mM nitrate. We conducted the incubation at an optical density of 0.1 and measured the results at a wavelength of 600 nanometers ($OD_{600}$). The measurement of nitrogenase activity involved the generation of nanomoles of ethylene per minute for each milligram of protein. We quantified the protein concentrations using a standard protein assay (Bio-Rad, Hercules, CA) three times, using bovine serum albumin (BSA) as the reference standard.

## Estimation of biofilm formation

After an overnight incubation, we performed the crystal violet experiment to quantify biofilm formation by washing the A1501 culture with K-medium. After allowing it to dry, we stained it with a crystal violet solution. After a set time, the extra dye was washed away, and the remaining crystal violet was dissolved to evaluate the optical density at 590 nanometers, which helped measure the amount of biofilm present. After adjusting the $OD_{600}$ to 0.5, 150 μL of a washed culture was added to the corresponding media in 96-well PVC plates (Corning Co., New York, NY, USA). The plates were then positioned in a growth-promoting environment at a temperature of 30°C and left undisturbed for 48 hours by measuring the optical density at 600 nanometers ($OD_{600}$ nm). We washed the wells three times with double-distilled water ($ddH_2O$) to measure the biomass of the biofilm. Each well received 160 μL of a 0.1% crystal violet solution and was incubated for 10 minutes. The wells were rinsed with double-distilled water ($ddH_2O$) repeatedly until the water was devoid of any purple hue. The crystal violet that stuck to the biofilm was removed using 30% acetic acid, and we checked how intense the color was at 540 nm with FixStation 3 (Molecular Devices, USA). We repeated the identical experiments to observe the biofilm. The biofilm was treated with violet. Relative quantitative polymerase chain reaction (qPCR) analysis was performed at different oxygen concentrations (0.5%, 1.0%, and 1.5%).

## Quantitative real-time PCR analysis

The quantitative RT-PCR analyses were performed following the manufacturer's directions, utilizing the ABI PRISM 7200 real-time apparatus from Applied Biosystems. The results were analyzed utilizing the ABI PRISM 7500 Sequence Detection System Software, created by Applied Biosystems. RNA isolation was conducted utilizing the Analytik Jena Kit and a qPCR thermocycler. We conducted cDNA synthesis using the TaKaRa Prime Script RT reagent kit, which includes a gDNA eraser to ensure accuracy in real-time applications. The table lists the primers used for qRT-PCR S3 Table.

## Bioinformatics analysis

The Virtual Footprint tool in the PRODORIC software (http://prodoric.tu-bs.de) was used to determine Anr regulon of *P. stutzeri* A1501 [20] S3 Fig. A phylogenetic tree was constructed using MEGA 7. We analyzed the evolutionary relationships between the Anr gene and its homologs in related species to understand its functional conservation. This analysis offered information about the regulatory mechanisms governing anaerobic responses in various Pseudomonas species. We used 1000 bootstrap iterations to arrive at a result of 0. We predicted the Anr protein and RpoS 3D protein structures using SWISS-MODEL (https://swissmodel.expasy.org/) [21], which provided the amino acid sequence. We identified putative active sites by entering the PDB structures into SPIDER (http://sppider.cchmc.org/) [22]. After finding the potential active site, both the Anr 3D protein and the RpoS protein were uploaded to Cluspro for molecular docking analysis. This step was to clarify how the Anr protein and the RpoS interact, how they might bind, and what that means for their functions. The interaction between Anr and RpoS was predicted on the ClusPro website (https://cluspro.org/login.php) [23] S4 Table.

### Western blot analysis for Anr and RpoS expression

We conducted Western blotting on protein extracts from bacterial cells cultured for 5 hours under nitrogen-fixing conditions. Proteins resolved by SDS/PAGE were transferred to a PVDF membrane. The membrane was then probed with specific antibodies against Anr and RpoS to detect their expression levels. We used a special light-producing substance to see how much protein was present under the nitrogen-fixing conditions. We performed electroblotting onto an Amersham membrane. We identified Anr and RpoS polypeptides using antisera against both proteins. Western blotting was performed on protein samples from bacteria grown for 12 hours in a biofilm environment to check for these proteins. The results showed a notable rise in the amounts of Anr and RpoS, indicating that the biofilm conditions successfully boosted their production in the bacterial cells. We maintained an oxygen tension of 5%. The cell pellet from 10-mL cultures with an $OD_{600}$ of 0.1 was mixed with 200 μL of SDS gel-loading buffer, heated for 10 minutes, frozen, and then spun in a centrifuge for 2 minutes. Subsequently, 20 μL of supernatants were applied to the stacking gel. Proteins were separated using a technique called SDS polyacrylamide gel electrophoresis (SDS/PAGE), which involved a special gel made from acrylamide and bis-acrylamide, and then transferred onto a PVDF membrane using a method called electroblotting. The membrane was left to sit with antibodies that target the Anr protein for 2 hours at 37 °C, then washed three times with TBS/Tween before being treated with anti-rabbit secondary antibodies for another 2 hours. We conducted the detection using the HRP-DAB Chemistry Kit (Tiangen).

### Statistical method

T-tests were utilized to determine the statistical significance of the colony counts observed on the selective and non-selective plates during the stability test. The experiment was conducted using GraphPad Prism version 7 for Windows, developed by GraphPad Software in San Diego, California, USA. S4 Table.

## Results

### Growth curve analysis

The growth of the *Pseudomonas* strain was observed to evaluate its metabolic capacity in both LB and a minimum medium (medium-K). Fig 2(A and B) shows that there was no noticeable alteration in the growth pattern of mutants and complementary strains. This evidence demonstrates that all mutant organisms may readily develop under conditions of limited nutrition supply without any interruption. Due to the oxygen sensitivity of the *anr* gene, it was necessary to monitor its growth under normal conditions. Any effects on the genomic level could potentially result in abnormal growth. The discrepancy in alteration could have indicated the potential incapacity of oxygen uptake in the system. The oxygen permeability of the system facilitates the efficient operation of all metabolic reactions taking place within it. The growth curve study of the mutant strain and complementary strain shows that there is no difference in the number of cell generations and how quickly they double when using minimal media.

### Effect of different oxygen concentration on *anr*

The complicated relationship between oxygen concentration, nitrogenase activity, and the regulatory role of the *anr* gene in *P. stutzeri* A1501 presents a fascinating area of study, especially considering the bacterium's metabolic versatility in diverse environments [24]. *P. stutzeri* A1501, known for its ability to remove nitrogen, has a complicated set of genes and enzymes that help it adjust to different oxygen levels, allowing it to live in both oxygen-rich and oxygen-poor environments. Understanding the regulatory mechanisms that govern nitrogen fixation and denitrification in response to oxygen availability is important for comprehending the bacterium's ecological role and potential applications in nitrogen cycling and bioremediation. Nitrogenase, the enzyme responsible for biological nitrogen fixation, is notably oxygen-sensitive, necessitating intricate protective mechanisms in aerobic or facultative anaerobic bacteria like *P. stutzeri* A1501 [25]. These

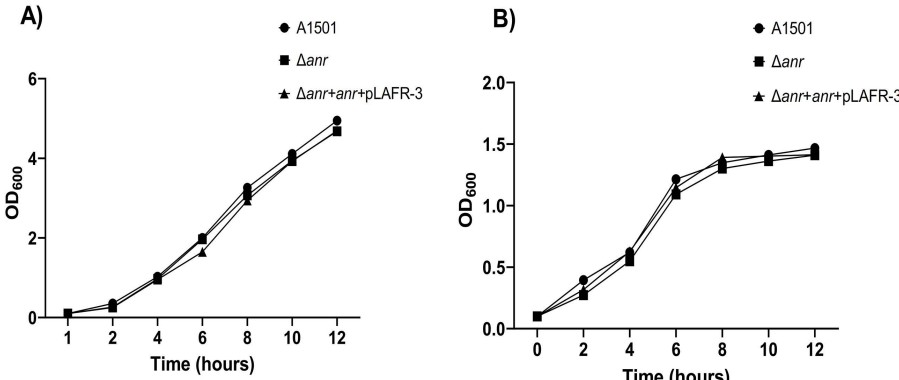

**Fig 2.** A) Growth pattern of the A1501 wild type Δ*anr*, and complementary strain, in LB- medium. B) Growth pattern of the A1501 wild type Δ*anr*, and complementary strain in K- medium. The growth in both the different media had no difference in comparison to the wild type.

mechanisms can include changes in behavior, physical barriers, and adjustments in body functions and chemistry, often working together to keep nitrogenase active. Oxygen is important for making energy through respiration, but it can seriously damage nitrogenase by permanently inactivating the enzyme. Working together in *P. stutzeri* A1501 shows how bacteria cleverly adapt to changing environments.

The capacity of *P. stutzeri* A1501 to modulate nitrogenase activity in response to varying oxygen concentrations is intricately linked to the regulatory functions of the *anr* gene. The Anr protein does more than just detect oxygen levels; it also helps control the activity of several genes that are important for adapting to low oxygen conditions and managing nitrogen processes. Anaerobic metabolism, including denitrification, is under the purview of Anr, which activates the expression of genes encoding enzymes responsible for reducing nitrate and nitrite to gaseous forms of nitrogen. Researchers are still exploring exactly how Anr interacts with the genes involved in nitrogen fixation and denitrification, but it's clear that this regulatory protein is vital for the bacterium to survive in areas where oxygen levels change. The activity of denitrification enzymes is adjusted, affecting the amounts of NO and $N_2O$ released during denitrification, which is important for lowering greenhouse gas emissions from wastewater treatment systems [26].

When looking at the A1501 under different nitrogenase activity conditions, it was found that the highest nitrogenase activity occurred at an oxygen level of 1%, as shown in **Fig 3**. The oxygen content is regarded as a crucial factor in assessing the biofilm that develops on the interface between liquid and air. The absence of the *anr* mutant, a transcriptional factor, resulted in a lack of reactive oxidative metabolism. This experiment was conducted to examine the alteration in gene expression caused by varying concentrations of oxygen (0.5%, 1.0%, and 1.5%) in a system containing *anr* genes. Gas-form oxygen was injected. Furthermore, the relative qPCR analysis revealed an increase in the expression of the *anr* gene at 1% oxygen. The mutation of these genes will provide us with an additional understanding of how it impacts nitrogenase activity and biofilm development.

## Oxidative stress

*P. stutzeri* A1501, a multifaceted bacterium renowned for its metabolic prowess, exhibits a remarkable capacity to acclimatize to various environmental conditions, particularly those characterized by fluctuating oxygen levels. This adaptability relies partly on a complex system controlled by the anaerobic regulator, a key protein that manages the genes needed for survival and functioning without oxygen. The Anr protein, which is part of a group of regulators that control gene activity, acts like a switch that detects changes in oxygen levels and adjusts gene expression accordingly. Under aerobic conditions, Anr remains inactive, allowing the bacteria to thrive by using oxygen as the primary electron acceptor in

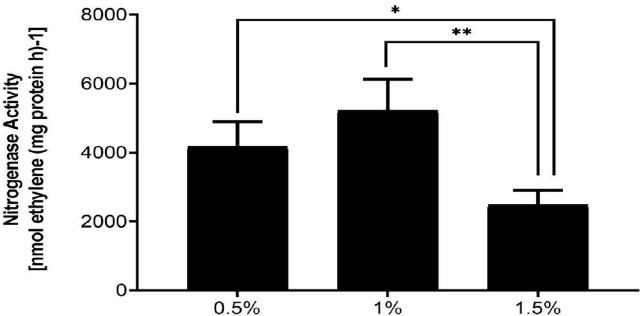

**Fig 3. Showing effect of oxygen on the percentage nitrogenase activity with different oxygen concentrations.** Among the 3 concentrations of oxygen which are 0.5%, 1% and 1.5% the highest nitrogenase activity was observed in 1%.

respiration [25,27,28]. As oxygen levels diminish, Anr undergoes a conformational change, enabling it to bind to particular DNA sequences in the promoter regions of its target genes [29]. Upon binding, Anr can either enhance or suppress the expression of specific genes, depending on the gene in question and the surrounding environmental conditions regulatory mechanism is crucial for the survival and ecological success of *P. stutzeri* A1501 in environments with fluctuating oxygen levels, including plant rhizospheres and anoxic aquatic sediments.

The oxidative role of Anr in *P. stutzeri* A1501 is complex, as it involves more than just its direct control of genes associated with anaerobic metabolism. Anr is mainly known for turning on genes that are important for anaerobic respiration, fermentation, and denitrification; however, people are starting to realize its role in helping the cell deal with oxidative stress and maintain a balance of redox reactions [3]. Under microaerobic conditions, Anr governs the expression of genes encoding enzymes that mitigate oxidative damage, such as superoxide dismutases and catalases. Superoxide dismutases convert superoxide radicals into hydrogen peroxide and oxygen, whereas catalases break down hydrogen peroxide into water and oxygen, preventing the buildup of these reactive oxygen species and protecting the cell from oxidative damage. The interplay between Anr and oxidative stress responses is crucial for *P. stutzeri* A1501's ability to thrive in habitats with restricted oxygen, as these conditions often lead to the generation of reactive oxygen species due to the partial decrease of oxygen. The presence of three separate nitrate reductases facilitates the exploration of the physiological importance of nitrate reduction in many environmental settings, especially regarding the roles of several systems in rhizosphere adaptation [30]. The ability of ancient nitric oxide reductases, cytochrome c oxidases, and quinol oxidases to bind and convert nitric oxide into nitrous oxide underscores their significance [31].

In addition to modulating antioxidant enzyme expression, *anr* also impacts the expression of genes that preserve cellular redox balance. This includes genes that encode enzymes responsible for controlling the levels of reducing equivalents, such as NADH and NADPH, which are essential for several metabolic pathways. Nitrate reductase particularly influences the expression of these genes to maintain a balanced redox state, thereby preventing excessive oxidation or reduction of cellular components. This is especially vital in anaerobic environments, where the electron transport chain increases the risk of redox imbalance. The expression of nitrate reductases is regulated by factors including light signaling, circadian rhythms, carbon dioxide concentrations, abscisic acid signaling, ethylene signaling, cytokinin signaling, and nitrogen metabolites [32]. Moreover, *anr* influences the regulation of genes related to the synthesis and metabolism of suitable solutes, such as ectoine and glycine betaine. These compounds serve as osmoprotectants, aiding the bacterium in regulating osmotic stress, often associated with anaerobic environments.

The Anr protein participates in biofilm formation; however, nitric oxide has opposing effects on bacterial adherence and biofilm development. Nitric oxide can notably induce biofilm dispersion under certain conditions, acting as a signaling molecule that triggers the detachment of cells from the biofilm matrix, nitric oxide, created during nitrate assimilation, can

block nitrate reductase via S-nitrosylation, demonstrating the connection of these systems [33]. This regulatory cascade allows *P. stutzeri* A1501 to accurately modify its metabolic processes in response to varying environmental conditions, hence improving its growth and survival.

The analysis of Anr's oxidative function in *P. stutzeri* A1501 is crucial for understanding the bacterium's ecological dynamics and its potential applications in bioremediation and biotechnology. Given that certain microbes can develop sophisticated molecular tactics against these detrimental agents, it is crucial to understand the oxidative mechanisms involved. By figuring out how *anr* controls gene expression based on oxygen levels and oxidative stress, researchers can learn how the bacterium adapts to different environments and its ability to break down pollutants or create useful chemicals. Understanding how Anr influences the expression of genes related to denitrification may assist in developing techniques that enhance the removal of nitrate from contaminated water sources. Furthermore, investigating Anr's role in regulating antioxidant enzyme expression may provide insights for the development of novel antioxidant strategies applicable in many biotechnological settings.

No. In this experiment, we observed sensitivity when we paired the moderate concentration of $H_2O_2$ with catalase activity. The *anr* gene showed no statistically significant change when exposed to oxidative stress from a 12 mM dose of $H_2O_2$ for 10 minutes. Findings indicate that elevated oxygen levels do not affect the function of the *anr* gene. S4 Fig.

## Nitrogenase activity

Examining the complicated relationship between nitrogenase and Anr control requires a comprehensive analysis of the environmental stimuli that initiate these processes, along with the fundamental molecular mechanisms involved. Nitrogenase, a complex metalloenzyme, is primarily linked to nitrogen fixation, the process of transforming atmospheric nitrogen into ammonia, a biologically accessible form for living organisms. Although oxygen typically hinders nitrogen fixation, we should not dismiss the possibility of nitrogenase contributing to denitrification under certain conditions, especially in its absence. Further investigation revealed multiple nitrate reductases, allowing researchers to study the importance of nitrate reduction in different environments, especially how different systems affect plant root health [24]. The activation of the Anr protein significantly elevates the expression of denitrification genes in anaerobic circumstances. This activation depends on the lack of oxygen and the availability of other electron acceptors, such as nitrate. To understand how nitrogenase and Anr work together, we need to look closely at the specific regulatory parts and binding sites in the promoter regions of nitrogenase genes.

Because nitrogenase is sensitive to oxygen, diazotrophs have developed ways to limit its exposure to oxygen; these ways include living in places without oxygen, creating barriers to stop oxygen from getting in, reducing oxygen levels around nitrogenase through their metabolism, and changing nitrogenase to make it more resistant to being inactivated [34]. The widespread occurrence of DNA and ANR genes in many microbes shows how important they are for maintaining usable inorganic nitrogen in ecosystems. In oxygen-deficient microenvironments within biofilms or soil aggregates, nitrogenase may demonstrate residual activity, influencing the complicated relationship between nitrogen fixation and denitrification processes. Uncertainty exists regarding the precise mechanisms governing this connection and the conditions under which nitrogenase can either directly influence or influence the Anr regulatory network. The regulation of denitrification in *P. stutzeri* A1501 is affected by several factors, including how much oxygen is present, the levels of nitrogen oxides, and the state of the cell's redox balance [35]. Anr serves as a pivotal hub, assimilating various signals to precisely regulate the expression of denitrification genes.

To understand how nitrogenase and Anr work together, we need to identify specific regulatory elements and binding sites in the promoter regions of nitrogenase genes. The complex link between nitrogenase and the Anr regulatory network in *P. stutzeri* A1501 poses an intriguing challenge. Looking into this connection will help us better understand how bacteria process nitrogen and show the important role of *P. stutzeri* in different environments, since it is involved in both nitrogen fixation and denitrification. The capacity to absorb nitrogen was formerly a metabolic specialization limited to a select set

of organisms; nevertheless, it is now recognized that this ability is prevalent among several bacterial species [36]. Several environmental parameters, including carbon availability, redox potential, pH, and salinity, influence the interaction between denitrification and DNA [9,37,38].

Environmental signals, such as ammonium and oxygen, regulate the expression of nitrogen fixation (nif) genes, contingent upon transcriptional activators. *P. stutzeri* A1501 may perform nitrogen fixation in the medium under low oxygen conditions. We noticed the differences in how nitrogen-fixing genes were expressed in the *anr* mutant compared to its normal cells. The results at a 1% oxygen level showed a big drop in nitrogen fixation in the *anr* insertion mutant. Fig 4(A).

## Relative qRT-PCR analysis for nitrogenase activity

A qPCR study of nitrogenase activity in the Δ*anr* strain showed that the levels of the *gacA, nifA, nifH,* and *ntrC* genes were lower. (See Fig 4(B). Anr helps to boost the activity of certain genes that are important for using different electron acceptors when there is little or no oxygen. Secondary transcription factors, activated in the presence of their specific electron acceptors, control the expression of these target genes. The anr mutant exhibits a loss of the capacity to thrive in anaerobic conditions. Thus, in a microaerophilic situation, the organism endeavors to maximize its oxygen consumption to ensure its survival in that particular environment. Nitrogenase activity is dependent on microaerophilic conditions. The absence of the *anr* gene significantly impacts the organism's essential metabolic pathways, leading to a decrease in nitrogenase activity and biofilm formation.

## Studying biofilm dispersal of *anr* genes

**Biofilm formation.**  The creation of biofilm in *P. stutzeri* A1501 is a complicated process influenced by many factors, with the anaerobic regulator playing a key role in this complex behavior [39]. Biofilms are a complex way for bacteria to adapt, allowing them to move from being free-floating to forming a group attached to a surface, which helps them survive better in different environments. The shift from free-floating bacteria to a group attached to a surface is guided by a combination of genetic and environmental signals, which results in a protective layer around the bacteria that forms the structure of the biofilm. The process of biofilm development happens in several stages, starting with the bacteria sticking to a surface, then forming small groups, growing into a mature biofilm, and finally spreading out, with each stage

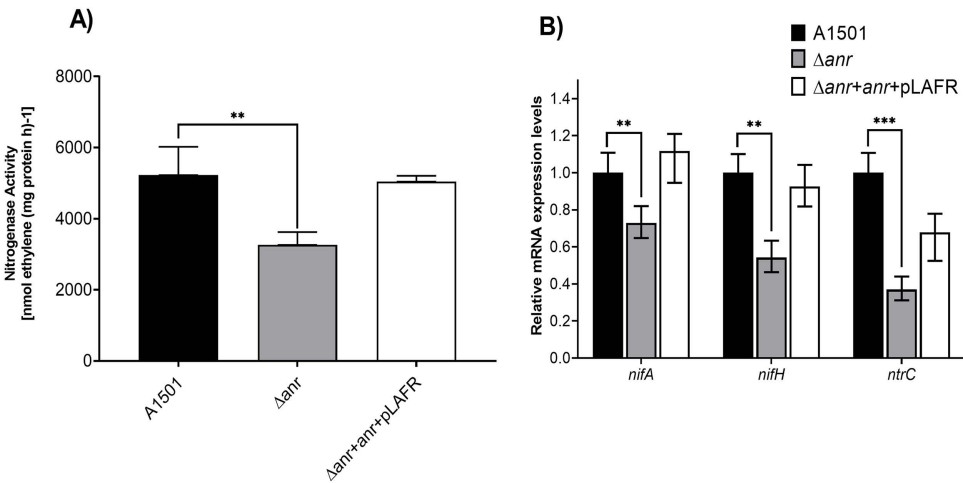

**Fig 4.  A) Nitrogenase Activity** A) Nitrogenase activity Δ*anr* having significantly lower nitrogenase activity in contrast to the wild type and after pLAFR with the *anr* gene retrieving the activity back to normal. B) Relative qPCR analysis showing Δ*anr* expressional levels at different oxygen concentrations.

controlled by specific processes. The anaerobic regulator, Anr, a crucial transcription factor in *P. stutzeri* A1501, reacts to fluctuations in oxygen levels and regulates the expression of several genes associated with diverse cellular functions and biofilm formation.

Anr's impact on biofilm development is complex, involving both direct and indirect regulation mechanisms [40]. An RNA can directly bind to the starting points of genes that build the biofilm, such as exopolysaccharides and adhesins, and control how much of these substances are made. Additionally, ANR can affect biofilm formation by changing the activity of genes involved in quorum sensing, which is a way for cells to communicate with each other to help organize and develop the biofilm. The regulation of gene and protein expression allows biofilm cells to rapidly acclimate to severe environmental conditions, as demonstrated by the increased production of multidrug efflux pumps and reduced cell permeability [41]. Ana's ability to sense and respond to changes in oxygen levels within the biofilm helps *P. stutzeri* A1501 improve its energy use and resource management, which strengthens the biofilm's structure and durability. As the biofilm layers proliferate, gradients of nutrients and oxygen develop, which results in phenotypic and metabolic diversity among the bacterial community [42]. The close proximity of cells in biofilms promotes the interchange of genetic material, potentially expediting adaptation and resistance to antimicrobial drugs [43].

Biofilm formation becomes more complicated because other proteins and signaling systems work with Anr to adjust gene activity and manage how cells function together. The production of exopolysaccharides, a key part of the biofilm structure, is controlled by Anr, which changes the activity of genes involved in making and moving them. Additionally, Anr adjusts the activity of genes that make flagella and pili, which are important for how bacteria move and stick to surfaces. The glycocalyx matrix, along with the ways bacteria pump out substances and certain enzymes, creates a protective layer that blocks harmful compounds and protects the outside of the biofilm [44]. Environmental conditions, such as nutrition supply, temperature, and pH, influence Anr's regulation of biofilm development. Biofilms account for a considerable percentage of enduring nosocomial infections, placing substantial strain on healthcare systems due to extended hospital stays and heightened treatment expenses [6,45]. The matrix constituents obstruct antibiotic infiltration and create a physical barrier against immune cells, exacerbating the persistence of biofilm-related illnesses. Biofilm-associated bacteria demonstrate heightened resistance to antibiotics relative to their planktonic counterparts, attributable to restricted drug diffusion, physiological alterations, and the existence of persister cells [46]. The extracellular matrix around bacterial cells in a biofilm acts as a major barrier to antimicrobial drugs and the body's immune responses. This matrix, consisting of extracellular polymeric components such as polysaccharides, proteins, eDNA, and lipids, establishes a protective milieu that facilitates bacterial survival and persistence [47,48]. The biofilm architecture obstructs antibiotic infiltration, inhibits bactericidal concentrations, and induces metabolic conditions that promote antibiotic tolerance [49–51]. The Anr-mediated regulation of biofilm formation in *P. stutzeri* A1501 illustrates the value of comprehending the complex regulatory networks that control bacterial adaptation and survival in various settings [49].

The Anr protein acts as a key controller, directly affecting the genes that help bacteria stick to surfaces, produce the materials needed for biofilm, and communicate with each other, all of which are important for building and maintaining the biofilm structure. Focusing on the Anr protein and its related factors could lead to effective ways to stop biofilm formation and tackle long-lasting infections. Biofilms are essential for human health, especially in long-term illnesses like cystic fibrosis, endocarditis, and chronic wound infections, which often do not respond well to regular antibiotic treatments. Looking into new ways to prevent and treat biofilms offers chances to reduce their negative effects in healthcare and industry.

Biofilms use a flexible defense system that helps bacteria survive against environmental challenges and medicines that fight infections. Biofilms enhance antibacterial resistance by obstructing antibiotic penetration [52]. Disrupting bacterial biofilms is acknowledged as a significant difficulty in clinical and industrial settings [45,47]. Targeting bacterial communication networks offers a novel strategy to mitigate biofilm formation [53]. Innovative approaches designed to inhibit biofilm development may improve the effectiveness of traditional antimicrobial drugs [54]. Often, conventional methods are insufficient for eliminating biofilms [55]. Therefore, it's important to create new ways to stop biofilms from forming or to get

rid of the biofilms that are already there. The significant recalcitrance in biofilm populations arises from multiple molecular mechanisms [56]. This includes how antimicrobials interact with the materials in the biofilm, slower growth rates, and genetic traits that affect resistance to antibiotics.

The organism experiences change due to the significant environmental modification in the soil. Particularly, the development of compounds that shield the organism from abrupt environmental shifts is crucial. These alterations are crucial for the organism's existence. A survival strategy entails the formation of biofilm in reaction to alterations in the soil environment. Bacterial aggregates encased inside a matrix exist in diverse environmental and biological settings. The cells constituting these communities possess distinct advantages over their free-floating counterparts in terms of protection against physical and chemical stresses. Biofilm bacteria demonstrate enhanced resistance to antimicrobial agents and immune system detection. A correlation between biofilm production and anaerobic metabolism has been established in the advanced phases of biofilm growth. Biofilms may experience constraints in oxygen availability, and they can detect fluctuations in oxygen levels. The cellular mechanisms that are responsible for detecting and responding to oxygen levels are complex and not yet fully understood. Oxygen-depleted environments, such as those found in biofilm and soil communities, promote increased anaerobic respiration *anr* activity. Pseudomonas species have the capability to sense and respond to diminished oxygen levels in their surroundings by utilizing a transcription factor known as *anr*. This transcription factor modulates transcription via a specific promoter, as elucidated by [57,58]. Under low oxygen conditions, these particular genes are activated, initiating a series of events that culminate in the formation of biofilms in anaerobic environments.

The biofilm formation was assessed using the 96-well plate method under the designated minimal conditions. The findings demonstrated that the Δ*anr* single mutant showed a reduction in biofilm development of up to 33.52% **Fig 5(A)**. The experiment entailed altering the oxygen levels within the system to investigate biofilm formation. We subsequently performed an expression analysis using relative qRT-PCR.

**Relative qPCR analysis for biofilm formation.** Anr, the homologue of Fnr in *E. coli*, demonstrates a significant association with the *rsm* non-coding RNA. Now we believe that non-coding RNA (ncRNA) molecules, known as *rsm*, significantly contribute to the positive regulation of RsmA in Pseudomonas species. Consequently, it is evident that the absence of the *anr* gene affects the regulation of non-coding RNA molecules. The small non-coding RNAs, known as Rsm, sequester and titrate RsmA to maintain the essential balance of this regulatory protein within bacterial cells. Conversely, RsmA possesses the capacity to modulate the synthesis of sRNA. The impact of low oxygen on the *rsm*X/Y/Z network was studied by looking at how the *rsm*X/Y/Z promoter worked in *P. stutzeri* A1501 grown in low-oxygen conditions. The results indicated a significant decrease in *gac*A expression levels in the Δ*anr* insertion mutant. A deficit in *gac*A leads to diminished or nonexistent expression of *rsm*XYZ, a vital regulator of biofilm growth. Anr appears to regulate the function of *nar*L within the system, as evidenced by the augmented activity of *nar*L. The illustration presented in **Fig 5(B)**.

## Bioinformatics analysis

The intricate regulatory network governing bacterial adaptation to diverse environmental conditions often involves complex protein-protein interactions, where transcriptional regulators play a pivotal role in orchestrating gene expression. In *P. stutzeri* A1501, Anr and RpoS are two key regulatory proteins that mediate responses to anaerobic stress and stationary phase, respectively, suggesting their importance in bacterial survival and adaptation. To elucidate the molecular mechanisms underlying their functions, computational approaches such as ClusPro, a widely used protein-protein docking server, can offer useful information about the potential interactions between Anr and RpoS and their target proteins, thus unraveling the regulatory circuitry of *P. stutzeri* A1501. The ClusPro web server facilitates protein-protein docking, a computational technique that predicts the three-dimensional structure of protein complexes. This approach allows for the exploration of various binding modes and the identification of energetically favorable interactions between proteins.

 

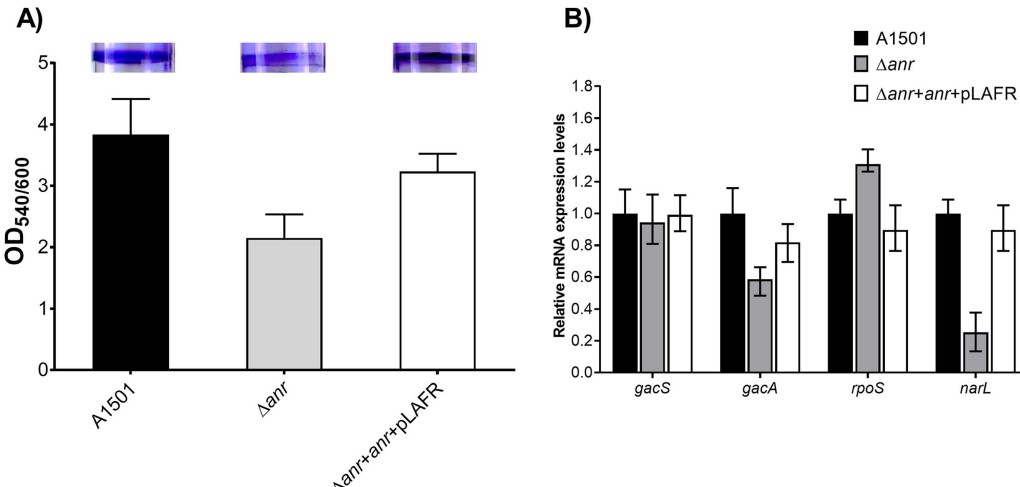

**Fig 5.** A) Biofilm formation observed in 96 well plate and the differential change that occurred in the surface biofilm formation after 48 hours of incubation was calculated. B) Biofilm formation at an oxygen concentration of 1% was observed for, Δ*anr*, and complementary cells. *gacA* was down-regulated but n*arL* was up-regulated during the biofilm formation.

Specifically, Anr, an anaerobic regulator, is crucial for enabling bacteria to survive in environments with limited oxygen availability, while RpoS, the sigma factor for stationary phase, orchestrates the expression of genes involved in stress resistance, nutrient scavenging, and biofilm formation, thus contributing to long-term survival and persistence [3]. Understanding the interplay between these two regulators is important when analyzing the adaptive strategies employed by *P. stutzeri* A1501 in fluctuating environments. By employing ClusPro, we can generate structural models of potential Anr-RpoS complexes and predict the amino acid residues involved in their interaction. This information can be used to design experiments to validate the interaction and to further investigate its functional consequences on the regulation of downstream target genes, potentially revealing novel mechanisms of gene regulation in *P. stutzeri* A1501. Such computational docking methods aid in identifying complex structures that are not immediately comprehensible. Predicting interactions between proteins is crucial for understanding cellular functions; however, a detailed analysis is necessary to fully grasp the chemical details, typically obtained through X-ray crystallography or nuclear magnetic resonance.

The ClusPro analysis can extend to investigate the interactions of Anr and RpoS with other proteins involved in regulatory or metabolic pathways. Investigating these interactions may also reveal novel regulatory connections and provide a more comprehensive understanding of the cellular processes governed by Anr and RpoS, offering a systems-level perspective on the adaptation of *P. stutzeri* A1501 to its environment. Integrating the ClusPro-derived structural models with experimental data, such as gene expression profiling and site-directed mutagenesis, will allow us to validate the predicted interactions and dissect their functional roles in regulating gene expression. For example, analyzing protein abundance changes during bacterial interactions can identify potentially regulated pathways [59]. Ultimately, this integrative approach will not only enhance our understanding of the molecular mechanisms governing bacterial adaptation but may also identify potential targets for developing novel antimicrobial strategies.

Using ClusPro to predict the structures of Anr and RpoS proteins in *P. stutzeri* A1501 could greatly enhance our understanding of how this bacterium functions and adapts to different environmental conditions. Anr, an anaerobic regulator, is crucial for regulating gene expression in response to oxygen availability, a critical adaptation for bacteria in fluctuating environments. RpoS, the sigma factor σS, is a key regulator of the overall stress response in bacteria and governs the expression of a broad regulon that promotes survival under various stress conditions, including nutrient deprivation, osmotic stress, and oxidative stress [60]. Obtaining accurate structural models of these critical regulatory proteins can

offer substantial understanding of their modes of action, including DNA binding, protein-protein interactions, and allosteric modulation. By combining predicted structures with additional experimental data like transcriptomics and proteomics, researchers can gain a complete understanding of the complicated regulatory systems that help bacteria adapt to their environment, which can help them create methods to change their behavior for biotechnological or bioremediation purposes. S5 Fig. The prediction between Anr and RpoS in **Table 3** shows the distance and hydrogen bonds linking the molecules.

The characterization of the *anr* promoter in *P. stutzeri* A1501, as predicted by Softberry, would not only shed light on the regulation of anaerobic respiration in this bacterium but also provide a valuable tool for biotechnological applications requiring oxygen-responsive gene expression [61]. Methods to study promoter activity commonly are based on the expression of a reporter gene from the promoter of the gene of interest [62]. Reporter assays provide only qualitative and semi-quantitative data about the spatial expression patterns of the linked promoter. Reporter genes can be used to monitor promoter activity, especially low-level transcription in vivo [63]. In addition, a strong synthetic promoter could be used to overexpress a natural type III polyketide synthase [64]. We identified the promoter using the website http://www.softberry.com/berry.phtml?subject=bprom&category=programs&subcategory=gfindb. The Anr to be shares the promoter region, with multiple genes in the *P.stutzeri* A1501 genome their sequences and initiation nucleotide distance are explained in the S5 Table. We were unable to further confirm the relationship between Anr and RpoS due to budget constraints. I hope more researchers will be able to achieve it in the future.

## Western blot analysis of Anr and RpoS expression

Western blot analysis, a fundamental method in molecular biology, offers a robust technique for examining protein expression and regulation across various biological systems. This method can be strategically utilized to analyze the functions of Anr (anaerobic regulator) and RpoS (sigma factor) in *P. stutzeri* A1501's response to environmental stimuli and cellular mechanisms [59]. We employ the Western blot technique to explore the expression of Anr and RpoS in *P. stutzeri* A1501. The examination of *P. stutzeri* A1501 under differing oxygen concentrations and stress circumstances facilitates a direct evaluation of their protein levels, elucidating the response of these crucial regulators to environmental cues. Comparing protein levels under varying conditions allows us to deduce the influence of environmental changes on the control of gene expression [65]. This methodology can yield significant insights into the mechanisms by which *P. stutzeri* A1501 acclimatizes to its environment.

The subsequent essential phase in Western blotting involves probing the membrane with specific antibodies that recognize Anr and RpoS. Primary We incubate specific antibodies, specifically designed to attach to target proteins, with the

**Table 3. Shows the interaction between Anr and RpoS using ClusPro.**

| Acceptor (Anr) | Donor (RpoS) | Hydrogen bond | Donor (Anr)-Acceptor (RpoS) Distance | Donor (Anr)-Hydrogen-Acceptor (RpoS) Distance |
|---|---|---|---|---|
| GLN 10.A NE2 | GLU 77.A OE2 | GLN 10.A HE21 | 2.909 | 1.940 |
| SER 19.A OG | ASP 138.A OD1 | SER 19.A HG | 2.820 | 1.962 |
| ARG 67.A NE | GLU 140.A OE1 | ARG 67.A HE | 2.913 | 1.982 |
| ARG 67.A NH2 | GLU 140.A OE2 | ARG 67.A HH22 | 2.721 | 1.824 |
| GLY 69.A N | GLY 142.A O | GLY 69.A H | 2.928 | 1.951 |
| LYS 107.A NZ | GLU 222.A OE1 | LYS 107.A HZ1 | 2.675 | 1.717 |
| ARG 140.A NH1 | ALA 2.A O | ARG 140.A HH11 | 2.654 | 1.755 |
| ARG 141.A NH1 | ARG 179.A O | ARG 141.A HH12 | 2.724 | 1.739 |
| ARG 154.A NE | GLN 185.A OE1 | ARG 154.A HE | 2.815 | 1.826 |
| SER 177.A OG | GLU 9.A OE1 | SER 177.A HG | 2.880 | 1.925 |
| ARG 227.A NH1 | GLN 155.A OE1 | ARG 227.A HH12 | 2.676 | 1.767 |

membrane to facilitate their preferential binding to Anr and RpoS. Subsequent to the removal of unbound primary antibodies, the membrane is treated with secondary antibodies conjugated to an enzyme, such as horseradish peroxidase or alkaline phosphatase. Secondary antibodies attach to primary antibodies, enhancing the signal and facilitating the detection of target proteins [66]. Chemiluminescent chemicals facilitate the visibility of protein bands [67]. The resultant signal is recorded using X-ray film or a digital imaging device, yielding a visual depiction of the protein bands [68]. The strength of these bands correlates with the quantity of Anr and RpoS proteins in each sample [68,69].

During SDS-PAGE, we successfully isolated the expression of distinct bands for the Anr and RpoS proteins after introducing the gene into *E. coli* using the pET28a plasmid tagged with hisx9. Fig 6(A and B). The Anr protein band detected was approximately 27.1 kDa. The RpoS protein band was seen at approximately 37.8 kDa. When utilized concurrently in *E. coli*, a band was observed at 64.9 kDa, indicating an interaction between RpoS and Anr protein. The separate plasmids were then put into *E. coli* together, and Western blotting was done to evaluate for protein interactions. As expected, the isolated band showed a larger size when tested with antiserum against Anr and RpoS, confirming that they interact. This indicates that in *P. stutzeri* A1501, RpoS and Anr interact to enhance the activation of downstream genes. Fig 6(C, D and E).

## Discussion

The intricate regulatory network that governs bacterial adaptation to fluctuating environmental conditions involves a sophisticated interplay of various transcription factors and signaling molecules, with Anr and RpoS playing pivotal roles in orchestrating these adaptive responses, particularly in metabolically versatile organisms such as *P. stutzeri* A1501. anr, an anaerobic regulator, functions as a global transcription factor that adjusts gene expression based on oxygen levels, allowing bacteria to thrive in both aerobic and anaerobic environments [13]. RpoS, the main controller of the general stress response, manages the activity of several genes related to handling stress, surviving tough conditions, and adapting to limited food and different environmental challenges. The synchronized function of Anr and RpoS facilitates the remarkable resilience and metabolic flexibility of *P. stutzeri* A1501, enabling its colonization of diverse ecological niches and survival against various environmental adversities. Additionally, the quorum sensing network in *P. aeruginosa* demonstrates adaptability, reacting to external biostress cues and allowing the pathogen to modulate virulence gene expression [70].

The study of nitrogenase activity in relation to the *anr* gene and different oxygen concentrations in *P. stutzeri* A1501 offers explanations for the intricate regulatory networks that govern bacterial adaptation to changing environmental conditions. Furthermore, the analysis of atypical variants in bacteria sheds light on the diverse ecophysiological roles and evolutionary adaptations within microbial communities [71]. The regulatory mechanisms governing nitrogen metabolism, particularly the role of Anr, are critical for understanding how bacteria such as *P. stutzeri* A1501 contribute to global nitrogen cycling. Understanding the complicated relationship among regulatory factors and environmental signals is essential for harnessing the potential of microorganisms to mitigate nitrogen pollution and enhance nitrogen use efficiency in various ecosystems [38]. In *Arabidopsis,* the activity of nitrate reductase is influenced by several factors, such as light, daily cycles, $CO_2$ levels, and different signaling molecules like ABA, ethylene, cytokinin, and nitrogen compounds. The Rnf complexes, found in various bacteria and archaea, exemplify how electron bifurcation is coupled to energy conservation [72]. A study of nitrogenase activity in the Δ*anr* strain revealed decreased levels of the genes *gac*A, *nif*A, *nif*H, and *ntr*C. Anr enhances the expression of specific genes crucial for utilizing various electron acceptors under conditions of low or absent oxygen. Secondary transcription factors, which are activated by their specific electron acceptors, regulate the expression of target genes. Biofilms serve as persistent reservoirs for pathogens in various environments, including industrial water systems, medical devices, and human tissues, significantly contributing to chronic diseases. Consequently, creative techniques are urgently needed to effectively tackle biofilm-associated illnesses, particularly those targeting the initial stages of biofilm formation or essential regulatory pathways [73]. For the development of innovative therapeutic strategies, it is crucial to understand the genetic basis of biofilm formation, specifically the role of the *anr* gene in *Pseudomonas stutzeri*

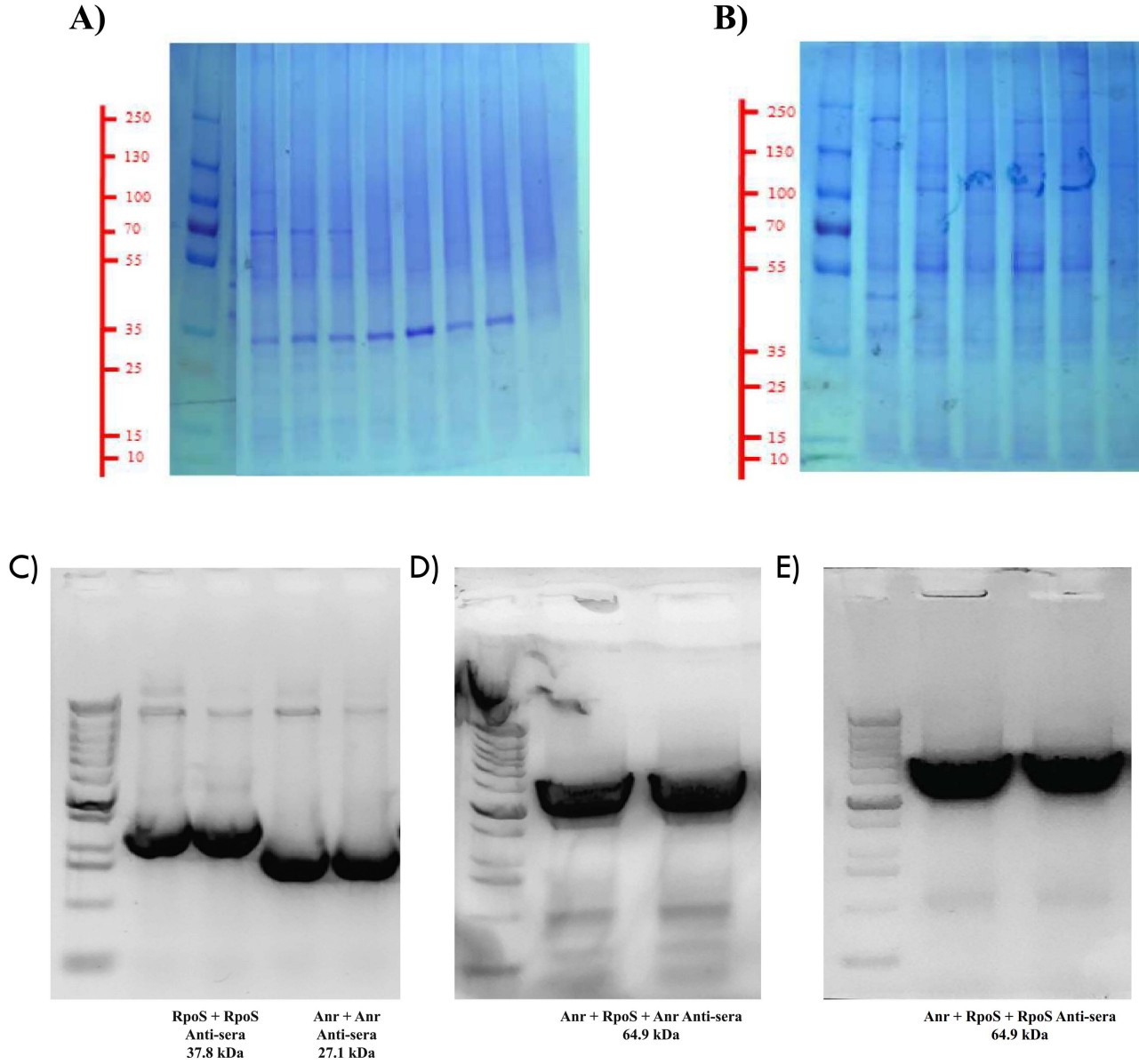

**Fig 6. SDS-PAGE A) SDS-PAGE Anr protein isolated in *E.coli* using pET28a plasmid.** The Anr protein band detected was approximately 27.1 kDa. B) SDS-PAGE RpoS protein isolated in *E.coli* using pET28a plasmid. The RpoS protein band was seen at approximately 37.8 kDa. C) Western Blotting Anr protein, and RpoS protein, D) Anr+RpoS protein using Anr anti sera, and E) Anr+RpoS protein using RpoS anti sera showing clear band. Band was observed at 64.9 kDa, indicating an interaction between RpoS and Anr protein.

A1501. The ability of bacteria to form biofilms is closely linked to the virulence of isolates from deep-seated infections, highlighting its crucial role in pathogenesis [74]. Thus, investigating the regulatory mechanisms, particularly those governed by the *anr* gene, that promote biofilm formation offers substantial prospects for the creation of customized antibiofilm therapies [75]. The Δ*anr* insertion mutant showed a markedly lower level of *gac*A expression, according to the data. A deficiency in *gac*A results in reduced or absent expression of *rsm*XYZ, an essential regulator of biofilm development. Anr seems to modulate the function of *nar*L within the system, as indicated by the significant decreased activity of *nar*L.

The Anr protein, a member of the CRP-FNR family of transcription factors, senses oxygen levels via its iron-sulfur cluster, which undergoes structural changes in the presence of oxygen, leading to adjustments in DNA binding affinity and subsequent regulation of gene expression. In *P. stutzeri* A1501, Anr regulates the expression of genes linked to anaerobic respiration, denitrification, and fermentation, allowing the bacterium to utilize alternative electron acceptors such as nitrate and nitrite in anoxic conditions [24]. *Anr* regulates the cellular redox balance and energy production during anaerobic growth, thereby enhancing the survival and proliferation of *P. stutzeri* A1501 in environments with limited oxygen. Investigations on equivalent organisms have revealed the importance of similar systems in responding to variations in oxygen and nitric oxide concentrations, highlighting the broader implications of these regulatory processes [76]. The cAMP receptor protein, a crucial transcriptional regulator, can influence numerous operons, with its effect on virulence demonstrated in Klebsiella pneumoniae, underscoring the significant role of such regulators in bacterial physiology [77].

RpoS, or σS, is an alternative sigma factor that directs RNA polymerase in the transcription of genes related to the general stress response. In *P. stutzeri* A1501, RpoS regulates the expression of genes linked to several stress resistance mechanisms, including oxidative damage, osmotic stress, and nutritional deprivation. RpoS modulates the expression of genes linked to biofilm formation, motility, and virulence, hence augmenting the bacterium's ability to survive and provoke infections in particular hosts. The expression and activity of RpoS are intricately controlled by a sophisticated network of regulatory components, such as short RNAs, proteases, and phosphorylation cascades, guaranteeing that the stress response is accurately synchronized and tailored to particular environmental conditions. The alternative sigma factor, encoded by sig-B, is present in Gram-positive bacteria such as *L. Monocytogenes* is influenced by environmental conditions and is essential for adaptation to heat, acid, and osmotic stressors [5]. The coordinated functioning of these systems illustrates the various response mechanisms available to bacteria in times of stress.

The interplay between Anr and RpoS in *P. stutzeri* A1501 is significant, as these regulators influence each other's expression and function, leading to complex regulatory networks and finely tuned adaptive responses. Anr may indirectly regulate RpoS expression by modulating the levels of particular metabolites or signaling molecules that affect RpoS stability or functionality [12]. Conversely, RpoS may influence Anr function by regulating the expression of genes related to oxygen metabolism or redox homeostasis, thereby affecting the cellular redox state and Anr's ability to sense oxygen availability. The integration of several regulatory signals allows *P. stutzeri* A1501 to prioritize different survival strategies according to prevailing environmental conditions, hence improving its chances for survival and proliferation. The intricate regulatory networks governing bacterial adaptation often encompass global regulators and stress-specific elements that meticulously modulate gene expression in response to environmental cues [78]. Understanding the complex regulatory pathways, especially those affected by short RNAs, is crucial for clarifying the mechanisms of biofilm formation and bacterial pathogenicity [39,65]. Further investigation is required to elucidate the molecular mechanisms of their regulation and their connections with other regulatory components. Investigating the specific DNA binding sites of Anr and RpoS, as well as their transcriptional targets, will offer substantial understanding of their roles in regulating gene expression in response to diverse environmental stressors. Furthermore, examining the post-translational modifications that affect Anr and RpoS activity, such as phosphorylation, acetylation, or proteolysis, will clarify the dynamic regulation of these proteins and their responses to cellular signaling networks. Further insight into regulatory pathways can be gained by analyzing DNA methylation sites in gene promoters, leading to altered transcription levels of several virulence genes, including a flagella gene, an RNA polymerase sigma factor, a capsular polysaccharide export protein, and a multidrug efflux pump.

## Patents

There is no patent resulting from the present work of this manuscript.

## Supporting information

**S1 Fig. Illustration showing the pk-mob-18 plasmid with a defective *anr* insertion in the genome of *P. stutzeri* A1501.**
(PDF)

**S2 Fig. Gel electrophoresis bands 1–2 amplified *anr* (up and down expression genes) from *P. stutzeri* A1501 wild type showing a single band and 3–4 amplified with pk-mob-18 with multiple bands confirming insertion.**
(PDF)

**S3 Fig. Promoter region of the *anr* gene with *ara*C initiation highlighted in bold.**
(PDF)

**S4 Fig. Oxidative stress test after exposure to 12mM of $H_2O_2$ for 10 minutes.** Their growth represents no significant change in the growth after 10-fold serial dilution between Δ*anr* and in wild type A1501.
(PDF)

**S5 Fig. Protein-Protein interaction between RpoS and Anr protein using Chimera, and using the website (**https://cluspro.org/login.php**). A)** Anr protein model **B)** RpoS protein model, **C)** A protein-protein interaction using clusPro web site predicted 29 models out of which model 26 had the lowest kinetic energy calculated which was −1113.1. **D)** Illustrating the position of interacting molecules in the globular structure of protein.
(PDF)

**S1 Table. Reagents used.**
(PDF)

**S2 Table. List of primers used.**
(PDF)

**S3 Table. List of qPCR primers used.**
(PDF)

**S4 Table. Software and algorithms.**
(PDF)

**S5 Table. Anr promoter with other genes from Virtual Footprinter (**https://www.prodoric.de/vfp/vfp_promoter.php**).**
(PDF)

## Acknowledgments

We extend our gratitude to Tang Xiaofang for her assistance and perceptive elucidation of this manuscript. I would like to thank Professor Dr. Lin Min of the Chinese Academy of Agriculture Science for providing the *P. stutzeri* A1501 strain. Dr. Faiz Muhammad helped in sharing the Anr and RpoS rabbit anti-sera.

## Author contributions

**Conceptualization:** Muhammad Ali Rasheed.

**Data curation:** Noureen Rasheed, Xize Ali.

**Formal analysis:** Shahab -ud-din, Arsalan Hassan.

**Software:** Irfan Khan, Muhammad Ali Rasheed.

**Validation:** Muhammad Ali Rasheed.

**Writing – original draft:** Muhammad Ali Rasheed.

**Writing – review & editing:** Muhammad Ali Rasheed.

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
