## [Decision Letter · Decision Letter 0]

20 Aug 2025

PONE-D-25-31925At low oxygen concentrations, the environmental impact of the anr gene on nitrogen fixation and biofilm formation in Pseudomonas stutzeri A1501PLOS ONE

Dear Dr. Rasheed,

Thank you for submitting your manuscript to PLOS ONE. After careful consideration, we feel that it has merit but does not fully meet PLOS ONE’s publication criteria as it currently stands. Therefore, we invite you to submit a revised version of the manuscript that addresses the points raised during the review process.

We look forward to receiving your revised manuscript.

Kind regards,

Naga Raju Maddela, Ph.D

Academic Editor

PLOS ONE

Journal Requirements:

3. Please remove your figures from within your manuscript file, leaving only the individual TIFF/EPS image files, uploaded separately. These will be automatically included in the reviewers’ PDF.

Reviewers' comments:

Reviewer's Responses to Questions

**Comments to the Author**

1. Is the manuscript technically sound, and do the data support the conclusions?

Reviewer #1: Yes

Reviewer #2: Yes

2. Has the statistical analysis been performed appropriately and rigorously? 

Reviewer #1: No

Reviewer #2: Yes

3. Have the authors made all data underlying the findings in their manuscript fully available?

Reviewer #1: Yes

Reviewer #2: Yes

4. Is the manuscript presented in an intelligible fashion and written in standard English?

Reviewer #1: No

Reviewer #2: No

5. Review Comments to the Author

Reviewer #1: General Comments:

1. Novelty and Significance:

The study provides valuable insights into the regulatory role of the anr gene in P. stutzeri A1501, especially its influence on nitrogen fixation and biofilm formation under microaerobic conditions. The connection between Anr, RpoS, and nitrogenase activity is novel and well explored.

2. Clarity and Structure:

The manuscript is rich in detail but suffers from redundancy and occasional repetition, particularly in the Results and Discussion sections. Some paragraphs would benefit from summarization for better flow and clarity.

3. Language and Grammar:

There are numerous grammatical issues and awkward phrasings throughout the manuscript. A professional language edit is recommended to improve readability and scientific clarity.

Major Comments:

The abstract provides a solid summary, but it is too lengthy and technically dense. Simplifying and shortening would improve impact.

The introduction is comprehensive, though some references (e.g., [1] to [7]) lack proper citation formatting. Consider shortening the background and focusing more clearly on the gap in knowledge addressed by this study.

Experimental protocols are clearly described, but, include strain source references directly in the tables instead of relying on supplementary figures alone. Better explain why certain oxygen concentrations (0.5%, 1.0%, 1.5%) were selected. The oxidative stress section lacks a positive control or a rationale for H₂O₂ dosage

While the results are rich in detail, they are narratively heavy. Breaking them into shorter paragraphs with subheadings would enhance readability. Figures mentioned (e.g., Fig 2, Fig 3, Fig 4A/B, Fig 5A/B) are not included in the main file. Ensure all figures are clearly labeled and discussed. Redundancy exists in the explanation of Anr’s regulatory role across nitrogen fixation and oxidative stress, these should be consolidated.

The discussion often reiterates findings from the results rather than providing critical analysis. Strengthen the comparative analysis with previous studies, especially for Pseudomonas spp. and biofilm formation pathways.

Consider discussing the limitations of the study more explicitly and suggesting future directions.

Western blot interpretation is sound, but quantification (e.g., densitometry values) is not clearly described. Bioinformatics docking results (Anr-RpoS interaction) are presented but need experimental validation or at least acknowledgment of limitations.

Other Minor comments

Define abbreviations (e.g., RpoS, qPCR) at first mention.

Avoid subjective phrases like “fascinating area of study” and instead use objective scientific language.

Use consistent tense throughout the manuscript—currently fluctuates between present and past.

Reviewer #2: The present work entitled “At low oxygen concentrations, the environmental impact of the anr gene on nitrogen fixation and biofilm formation in Pseudomonas stutzeri A1501” is significant work to undertake for research. The manuscript is clear and easy to understand. Even though, in the manuscript there are some minor mistakes as well as grammatical mistakes, hence the authors should be corrected and improve the language.

The precise comments specified below:

The authors should be followed journal format throughout the manuscript.

Revise the complete manuscript by native English speaker.

Line No:100- interaction should be interaction

Line No:108 – Pseudomonas should be Pseudomonas

Line No: 113 - Cellvibrio, Halomonas, and Rhodospirillum. Should be Cellvibrio, Halomonas, and Rhodospirillum.

Tables – weight (g)- need not mention 5 g 10 g 10 g etc., should be 5 10 10 etc.,

In text some places 30 degrees Celsius in some places 30 oC, it should be maintain uniform throughout the manuscript.

Line No:115 & 116: The suspension was agitated at 220 rpm, adhering to the methodology established by [12]- incomplete sentence.

Line No: 179 & 180 - We repeated the identical We repeated the identical experiments to observe the biofilm. The biofilm was treated with the biofilm was treated with violetRepeated the sentence.

Line No: 312 & 313- The observation establishes a correlation between the abundance of oxygen and an increase in ER expression. Why its in red colour?

6. PLOS authors have the option to publish the peer review history of their article (what does this mean? ). If published, this will include your full peer review and any attached files.

**Do you want your identity to be public for this peer review?** For information about this choice, including consent withdrawal, please see our Privacy Policy .

Reviewer #1: No

Reviewer #2: No

---

## [Author Response · Author response to Decision Letter 1]

2 Sep 2025

PONE-D-25-31925

At low oxygen concentrations, the environmental impact of the anr gene on nitrogen fixation and biofilm formation in Pseudomonas stutzeri A1501

PLOS ONE

Dear Naga Raju,

Thank you for your email. I am also thankful for the reviewer’s insightful information on the manuscript. Please find the detailed point by point response on the article as below in blue,

Answer: In the new manuscript the style is strictly followed as per the PLOS one guideline available on the above-mentioned website.

Answer: In Fig-6 the entire Western blot for the Anr and RpoS has been added. The pictures are uncropped and in 300dpi TiFF format. Additionally, we shall send the All the protein blots by email to plosone@plos.org for review.

3. Please remove your figures from within your manuscript file, leaving only the individual TIFF/EPS image files, uploaded separately. These will be automatically included in the reviewers’ PDF.

Answer: In Fig-6 the entire Western blot for the Anr and RpoS has been added. The pictures are uncropped and in 300dpi TIFF format. Additionally, we shall send the All the protein blots by email to plosone@plos.org for review.

Answer: All supporting materials captions are mentioned in the end of the manuscript as per the guideline of PLOS one information provided by the official web site.

Answer: The reviewer did not ask to add any previously published work in the review however; a previously published reference was added Reference number 26. It was presented in a seminar in 2019 at Shandong, China.

Reviewers' comments:

Reviewer's Responses to Questions

Comments to the Author

1. Is the manuscript technically sound, and do the data support the conclusions?

Reviewer #1: Yes

Reviewer #2: Yes

2. Has the statistical analysis been performed appropriately and rigorously?

Reviewer #1: No

Reviewer #2: Yes

3. Have the authors made all data underlying the findings in their manuscript fully available?

Reviewer #1: Yes

Reviewer #2: Yes

4. Is the manuscript presented in an intelligible fashion and written in standard English?

Reviewer #1: No

Reviewer #2: No

5. Review Comments to the Author

Reviewer #1: General Comments:

1. Novelty and Significance:

The study provides valuable insights into the regulatory role of the anr gene in P. stutzeri A1501, especially its influence on nitrogen fixation and biofilm formation under microaerobic conditions. The connection between Anr, RpoS, and nitrogenase activity is novel and well explored.

2. Clarity and Structure:

The manuscript is rich in detail but suffers from redundancy and occasional repetition, particularly in the Results and Discussion sections. Some paragraphs would benefit from summarization for better flow and clarity.

3. Language and Grammar:

There are numerous grammatical issues and awkward phrasings throughout the manuscript. A professional language edit is recommended to improve readability and scientific clarity.

Major Comments:

The abstract provides a solid summary, but it is too lengthy and technically dense. Simplifying and shortening would improve impact.

Answer: The abstract was not shortened as it might loss its importance to the study for the scientific audience. However small modification was done to make it more academic sound.

The introduction is comprehensive, though some references (e.g., [1] to [7]) lack proper citation formatting. Consider shortening the background and focusing more clearly on the gap in knowledge addressed by this study.

Answer: The references were updated to have relevant formatting as per requirement of PLOS one. The introduction is concise without changing the core focus of the study to build the foundation of it.

Experimental protocols are clearly described, but include strain source references directly in the tables instead of relying on supplementary figures alone. Better explain why certain oxygen concentrations (0.5%, 1.0%, 1.5%) were selected. The oxidative stress section lacks a positive control or a rationale for H₂O₂ dosage

Answer: Table (b) was added in the manuscript for explaining strains used in this study. The importance of oxygen in nitrogenase activity is explained from line 303 to 312. The different concentration shows that the oxygen Pseudomonas stutzeri A1501, oxygen concentration is not a simple. It is a key regulatory signal that determines the metabolic mode of the bacterium. Its ability to sense and respond to fine gradients of O₂ switching between aerobic respiration, denitrification, and nitrogen fixation is what makes it a highly successful and adaptable plant-associated bacterium in its natural environment. The rational for the dosage of the H2O2 is explained in Method and Material of Oxidative stress line number 195.

While the results are rich in detail, they are narratively heavy. Breaking them into shorter paragraphs with subheadings would enhance readability. Figures mentioned (e.g., Fig 2, Fig 3, Fig 4A/B, Fig 5A/B) are not included in the main file. Ensure all figures are clearly labeled and discussed. Redundancy exists in the explanation of Anr’s regulatory role across nitrogen fixation and oxidative stress, these should be consolidated.

Answer: Results are now focused and more concise and be easy for audience to understand. For clarification Figures are cited as per lines they are in the manuscript.

Figures Line number in Manuscripts

Fig-1 Line number 155

Fig-2 Line number 282

Fig-3 Line number 324

Fig-4 Line number 444 & 447

Fig-5 Line number 550 & 565

Fig-6 Line number 660

Anr does not have any particular role in oxidative stress however anr importance in nitrogen fixation condition is expressed manuscript Check from line 429 till 443. As oxygen is key player controlling regulation of both nif island genes and anr makes it relevant to study.

The discussion often reiterates findings from the results rather than providing critical analysis. Strengthen the comparative analysis with previous studies, especially for Pseudomonas spp. and biofilm formation pathways.

Consider discussing the limitations of the study more explicitly and suggesting future directions.

Answer: Discussion is now primarily focused on the work done. Understanding the genetic foundations of biofilm formation, particularly the function of the anr gene in Pseudomonas stutzeri A1501, is essential for the advancement of innovative therapeutic strategies. The capacity of bacteria to form biofilms is significantly associated with the virulence of isolates from deep-seated illnesses, underscoring its essential role in pathogenesis of other pseudomonas species. Consequently, examining the regulatory mechanisms, including those regulated by the anr gene, that facilitate biofilm formation presents significant opportunities for the development of tailored antibiofilm therapeutics. This necessity propels research into fundamental mechanisms such as quorum sensing, a cell density-dependent communication mechanism that regulates biofilm formation and resistance, signifying a prospective target for anti-biofilm strategies.

Western blot interpretation is sound, but quantification (e.g., densitometry values) is not clearly described. Bioinformatics docking results (Anr-RpoS interaction) are presented but need experimental validation or at least acknowledgment of limitations.

Answer: The manuscript provided an explanation of the western blot's density. Because of a lack of funds, the bioinformatics docking finding was not validated; this restriction was acknowledged in the publication and left for further research. Kindly check the line number 635.

Other Minor comments

Define abbreviations (e.g., RpoS, qPCR) at first mention.

Answer: RpoS and qPCR are abbreviated at line number 49 and 43 respectively. Other Abbreviations are also clearly defined in the manuscript.

Avoid subjective phrases like “fascinating area of study” and instead use objective scientific language.

Answer: All nonacademic statements were replaced with academic/scientific language from the manuscript.

Use consistent tense throughout the manuscript—currently fluctuates between present and past.

Answer: The entire manuscript was checked for grammatical errors. Hopefully now the flow of the manuscript will be found constant.

Reviewer #2: The present work entitled “At low oxygen concentrations, the environmental impact of the anr gene on nitrogen fixation and biofilm formation in Pseudomonas stutzeri A1501” is significant work to undertake for research. The manuscript is clear and easy to understand. Even though, in the manuscript there are some minor mistakes as well as grammatical mistakes, hence the authors should be corrected and improve the language.

The precise comments specified below:

The authors should be followed journal format throughout the manuscript.

Answer: The PLOS one format was strictly followed in the updated manuscript.

Revise the complete manuscript by native English speaker.

Answer: The entire manuscript was checked for grammatical errors. Hopefully now the flow of the manuscript will be found constant.

Line No:100- interaction should be interaction

Answer: Updated

Line No:108 – Pseudomonas should be Pseudomonas

Answer: Updated

Line No: 113 - Cellvibrio, Halomonas, and Rhodospirillum. Should be Cellvibrio, Halomonas, and Rhodospirillum.

Answer: Updated

Tables – weight (g)- need not mention 5 g 10 g 10 g etc., should be 5 10 10 etc.,

Answer: Updated

In text some places 30 degrees Celsius in some places 30 oC, it should be maintained uniform throughout the manuscript.

Answer: Updated, now single format is followed.

Line No:115 & 116: The suspension was agitated at 220 rpm, adhering to the methodology established by [12]- incomplete sentence.

Answer: Updated in Method and material section line number 206 in the new manuscript.

Line No: 179 & 180 - We repeated the identical We repeated the identical experiments to observe the biofilm. The biofilm was treated with the biofilm was treated with violet Repeated the sentence.

Answer: Updated

Line No: 312 & 313- The observation establishes a correlation between the abundance of oxygen and an increase in ER expression. Why it’s in red colour?

Answer: Updated and removed. It was a wrong statement which was removed.

6. PLOS authors have the option to publish the peer review history of their article (what does this mean?). If published, this will include your full peer review and any attached files.

Do you want your identity to be public for this peer review? For information about this choice, including consent withdrawal, please see our Privacy Policy.

Reviewer #1: No

Reviewer #2: No

Answer: Figures were modified using PACE digital diagnostic tool and were updated on the manuscript account along with that the figures were sent to PLOS using the Figure@plos.org. in Zip file downloaded from PACE.

Thank you so much again for your valuable insight on my ma

---

## [Editor Report · Decision Letter 1]

10 Sep 2025

The function of Anr in the differential effects of oxygen levels on biofilm development and nitrogenase performance in Pseudomonas stutzeri A1501

PONE-D-25-31925R1

Dear Dr. Rasheed,

We’re pleased to inform you that your manuscript has been judged scientifically suitable for publication and will be formally accepted for publication once it meets all outstanding technical requirements.

Kind regards,

Naga Raju Maddela, Ph.D

Academic Editor

PLOS ONE
---

## [Editor Report · Acceptance letter]

PONE-D-25-31925R1

PLOS ONE

Dear Dr. Rasheed,

I'm pleased to inform you that your manuscript has been deemed suitable for publication in PLOS ONE. Congratulations! Your manuscript is now being handed over to our production team.

Kind regards,

on behalf of

Dr. Naga Raju Maddela

Academic Editor

PLOS ONE